# Composition of Hybrid Deep Learning Model and Feature Optimization for Intrusion Detection System

**DOI:** 10.3390/s23020890

**Published:** 2023-01-12

**Authors:** Azriel Henry, Sunil Gautam, Samrat Khanna, Khaled Rabie, Thokozani Shongwe, Pronaya Bhattacharya, Bhisham Sharma, Subrata Chowdhury

**Affiliations:** 1Department of Computer Sciences and Engineering, Institute of Advanced Research, Gandhinagar 382426, Gujarat, India; 2Department of Computer Science and Engineering, Institute of Technology, Nirma University, Ahmedabad 382481, Gujarat, India; 3Department of Engineering, Manchester Metropolitan University, Manchester M1 5GD, UK; 4Department of Electrical and Electronic Engineering Technology, University of Johannesburg, Auckland Park, P.O. Box 524, Johannesburg 2006, South Africa; 5Department of Computer Science and Engineering, Amity School of Engineering and Technology, Amity University, Kolkata, 700135, West Bengal, India; 6Chitkara University Institute of Engineering and Technology, Chitkara University, Rajpura 140401, Punjab, India; 7Department of Masters of Computer Application, Sri Venkateswara College of Engineering and Technology (A), Chittoor 517127, Andhra Pradesh, India

**Keywords:** convolution neural network, IDS, deep learning, anomaly detection

## Abstract

Recently, with the massive growth of IoT devices, the attack surfaces have also intensified. Thus, cybersecurity has become a critical component to protect organizational boundaries. In networks, Intrusion Detection Systems (IDSs) are employed to raise critical flags during network management. One aspect is malicious traffic identification, where zero-day attack detection is a critical problem of study. Current approaches are aligned towards deep learning (DL) methods for IDSs, but the success of the DL mechanism depends on the feature learning process, which is an open challenge. Thus, in this paper, the authors propose a technique which combines both CNN, and GRU, where different CNN–GRU combination sequences are presented to optimize the network parameters. In the simulation, the authors used the CICIDS-2017 benchmark dataset and used metrics such as precision, recall, False Positive Rate (FPR), True Positive Rate (TRP), and other aligned metrics. The results suggest a significant improvement, where many network attacks are detected with an accuracy of 98.73%, and an FPR rate of 0.075. We also performed a comparative analysis with other existing techniques, and the obtained results indicate the efficacy of the proposed IDS scheme in real cybersecurity setups.

## 1. Introduction

Information technology is growing very rapidly throughout the world. It has made information exchange very smooth and easy. However, these advancements have created many challenges for the communication system. Such a system or network faces intrusions in the form of different attacks. An IDS is a tool that can classify or detect potential cyber attacks in the host/network by applying detection algorithms. There are two classes of IDS, Signature and Anomaly based IDSs (SIDS and AIDS). In SIDS, attacks are detected considering the pre-defined pattern/signature of the attacks. In the AIDS network, traffic patterns are monitored and compared with normal or regular patterns in the network to detect any intrusion. Any change in the network is declared as an intrusion in the network by the Anomaly based IDS. It has an advantage over SIDS as it can execute the detection of new attacks in the network. Whereas SIDS can only detect attacks that match the previously stored signatures. Moreover, for data sources, IDS can be divided into Network- and Host-based IDSs. By looking at the information from the OS, server and firewall, and database logs, or application system audit, HIDS can identify threats from within the system. NIDS can identify external attacks before they reach the computer network. To identify potential network dangers, NIDS tracks and examines network traffic that is gathered from various network data sources [1,2].

There are several approaches to developing SIDS and AIDS. AIDS can nullify the limitations of SIDS and researchers have shown more interest in it. First, AIDS can be created using schemes such as Machine-learning/deep-learning (ML/DL), knowledge-based or statistical-based methods [3]. The statistical method detects intrusion using a statistical system of measurements such as the mean, standard deviation, mode, and median. Univariate, Multivariate, and Time-series models are the models used to implement Statistical IDS. Second, models are developed using knowledge-based techniques based on a set of rules made using human knowledge. Methods to develop knowledge-based IDS involve description languages, expert systems, and finite-state systems. Thirdly, machine learning is an extensively used technique to develop an Anomaly based IDS. The ML approach comprises two classes namely, unsupervised, and supervised learning. In addition to these categories, there is hybrid learning called semi-supervised learning. In contrast to unsupervised learning, which needs unlabelled instances for training, supervised learning needs labelled instances. The semi-supervised technique utilizes fewer categorized instances and more uncategorized input instances for training [4,5].

Machine learning methods to detect cyber intrusions are being widely used because of their automatic and timely manner of action. However, cyber intrusions are continuously changing which creates the requirement for more scalable detection systems. The opportunity to create such scalable and adaptable detection systems is provided by DL techniques. DL can be used with the supervised as well as unsupervised method. Unsupervised algorithms are used to produce labels for unlabelled instances [6,7]. When using techniques such as machine learning/deep learning (ML/DL), the dataset is crucial. Various IDS datasets are available for developing an IDS such as KDD 99 dataset, Centre for Applied-Internet Data-Analysis (CAIDA) dataset, DARPA Lincoln Lap Packet trace, UNSW-NB15, Coburg University published dataset-CIDDS-001, or New Brunswick University published dataset-CICIDS 2017. It includes a wide variety of typical attacks in addition to benign ones. Attacks such as DoS/DDoS, Heartbleed, botnet, SSH bruteforce, infiltration, and web attacks are included in CICIDS 2017 dataset [8,9].

### 1.1. Novelty of Paper

We used a deep learning algorithm called CNN with GRU framework to develop an IDS. CNN is widely used as a deep learning technique in various fields. However, these methods have a problem with long-term dependencies. The existing deep learning solutions deal with this problem of long-term dependencies. When long-term interactions are provided exponentially decreasing weights, the long-term dependency problem occurs. Consequently, it becomes challenging for the model to track previous data. GRU can fix this issue and enhance the deep learning model. Hence to effectively detect the attacks we used the GRU framework with RNN.

### 1.2. Contribution of Paper

To effectively classify the attacks, we designed an IDS using a deep learning technique. We used CNN to detect the attacks in IDS. To overcome the limitations of the existing DL technique as discussed in the previous section, the technique is modified using the GRU framework. We tried different combinations of CNN layers and GRU sequences. However, we achieved better performance with three CNN layers and two GRU layers. The detailed model strategy is shown in the latter section. In addition, we optimized the CICIDS 2017 dataset for better performance. The optimization includes the removal of redundant features and the selection of distinguished features.

### 1.3. Organization of Paper

The remaining text is presented in the below order: Section 2 manifests the most recent research on this subject. The proposed method is showcased in Section 3. The main contribution of the research is also covered in this section and it discusses the proposed technique in detail. Section 4 analyses the model’s performance using a variety of parameters, i.e., findings and discussion. The work of this study comes to a close in Section 5.

## 2. Related Work

The scope to enhance the IDS is prevailing despite the numerous varieties of approaches to deal with the same. Recent years have seen an increase in the development of IDSs based on machine learning relative to previous methodologies. Nevertheless, there are various methods to develop an IDS which performs differently in various scenarios.

Roberto et al. [10] illustrated a model to detect network intrusions using four ML models namely Multinomial LR, SVC-L and SVC-RBF, and RF. They evaluated their models using the UGR’16 dataset. They considered four attacks namely e Botnet, DoS, Scan, and Spam. They assert that the scientific community will benefit from their findings by developing better NIDS solutions. However, this research reflects on only four attacks.

Amar et al. [11] demonstrated ML techniques such as KNN, Naïve Bayes, Logistic regression, and SVM. They used the NSL-KDD to check the effectiveness of their models. They claimed that for binary and multi-class classification, KNN is superior to NB, and RF SVM with a precision factor of 93.28% to above 99.4% and an accuracy of 96.69%.

Iram et al. [12] demonstrated a detailed study using machine learning algorithms namely SVM, KNN, LR, NB, MLP, RF, ETC, and DT. They randomly selected the features from the NSL-KDD to reduce the dimension of the dataset. They claimed that RF, extra-tree, and DT classifiers managed to achieve an accuracy score of above 99%. However, they did not focus on the sight of executing optimization methods. Abdulsalam et al. [13] showcased the use of DT, RF, and XGBoost to develop an IDS for software-defined networks (SDN). NSL-KDD was utilized to test the effectiveness of these models. XGBoost classifier outperformed other classifiers when evaluated using various metrics such as F1 score, precision, recall, etc. However, they stated that deep learning algorithms, such as Auto-Encoder, GANs, and RNNs, such as GRU and LSTM can be used to conveniently detect the changes in the network. Raisa et al. [14] demonstrated the analysis of the learning algorithms namely DT, GBT, AdaBoost, MLP, LSTM, and GRU for detecting the intrusions. They used UNSW-NB 15 and Network TON datasets to test these models. To optimize the dataset, they used an embedded technique called the GIWRF model. The decision tree outperformed other classifiers in this study. However, this study lacks the multiclass classification.

Achmad et al. [15] demonstrated a hybrid strategy that incorporates the feature optimization method, which stands for supervised method, and the data reduction method, which stands for unsupervised method. Attribute importance DT-based technique with recursive feature removal is used to pick pertinent and important attributes, and the LOF method is used to identify anomalous or outlier data. To check the effectiveness of their model they used NSL-KDD and UNSW-NB15. They claimed that their model increases accuracy of the system when compared with other existing models. They noted that their model needs improvement in terms of sensitivity, specificity, and FAR.

Gustavo et al. [16] introduced the AB-TRAP framework to facilitate the full deployment of the solution, which allows the use of new network traffic and takes operational factors into account. Their methodology includes developing attack and legitimate datasets, training machine learning models, putting the solution into practice on a target system, and assessing the effectiveness of the security module. They demonstrated the performance of the ML models such as KNN, RF, XGB, NB, DT, MLP, SVM, and LR. They claimed that the decision tree (DT) provides the best result compared with other models. However, their model needs to be tested on the standard datasets containing a variety of attacks.

Maonan et al. [17] proposed a method to enhance interpretation of IDSs, this approach employs Shapley Additive Explanations (SHAPs) and integrates local and global explanations. The local explanations provide the justifications for the decisions the model makes in response to a particular input. The NSL-KDD dataset is utilized to test the framework’s viability. They used KNN, RF, SVM-RBF, one-vs.-all, and multiclass models as learning models. The experimental outcomes demonstrate that the interpretation outcomes produced by their framework are consistent with the traits of the attacks, and the outcomes are quite comprehensible.

Samson et al. [18] demonstrated the use of CNN to distinguish the attacks in IDS. To evaluate the model, they used CICIDS 2017 dataset. They managed to reach an accuracy rate of 94.96%. Moreover, they claimed that their model can detect new DoS instances other than the instances in the training phase. Priyanka et al. [19] showcased different IDS techniques based on CICIDS 2017 dataset. They divided the process into two categories, namely two class and multi-class for RF, NB, and Convolution Neural Network (CNN) techniques. They noticed that Nave Bayes had poor accuracy for two classes and had a lower multi-class performance. Moreover, RF has high accuracy compared with the CNN model. However, they used a partial dataset for evaluation. Sun et al. [20] proposed a method called CNN-LSTM. They modified the CNN model using LSTM for improving the detection rate. The model’s effectiveness was assessed using CICIDS 2017 dataset. Their result shows an accuracy of 98.7%. However, for a few attacks such as Heartbleed and SSH-Patator attacks the model showed a low detection accuracy rate.

Mario et al. [21] showcased the experiment-based comparison of neural-based techniques. The key focus of their work is on ANN. They used CICIDS2017/2018 and KDD99 datasets to evaluate their model. They claimed that the ANN-based techniques perform outstandingly in almost all cases, but as a result of the backpropagation technique they have the disadvantage of being slow. However, they lack the feature optimization step which can reduce time complexity of the classifier.

Shi et al. [22] discussed the use of the Semi-Supervised Deep Reinforcement method. The SSDDQN uses an auto encoder to recreate the attributes first, and a deep NN as a second step. NSL-KDD and AWID datasets were used in their experiment for testing and training. They claimed that for abnormal traffic their model achieved good results. However, the SSDDQN model’s optimization impact is constrained, and it has essentially no detection capacity for smallest number of U2R anomalous attack traffic.

Charlotte et al. [23] demonstrated the comparison of random forest with two deep learning algorithms: RNN and CNN. They carried out the experiment using Sentinel-2 time series. Analysing the algorithms, they claimed that CNN obtains the highest accuracy. Moreover, they also claimed that RNNs have more time complexity and less accuracy.

Joohwa et al. [24] presented an approach for deep learning classification using features that were extracted, not as a classification approach, but as a pre-processing technique for feature extraction. An unsupervised deep learning autoencoder model that is typically employed is classified by the Random Forest (RF) classification method, and features are extracted from that model using a deep sparse autoencoder. They used CICIDS 2017 dataset to perform the experiments. They claimed that the proposed approach was compared with existing methods for feature extraction and it was superior. However, the performance of the approach was somewhat poor for the rare class that existed in the network.

Mohammadnoor et al. [25] showcased a multi-stage optimised ML-based NIDS framework. They examined how oversampling approaches affect the size of the training instances for models and establishes smallest training sample size. They compared gain and correlation-based techniques for feature selection. They used two datasets to evaluate their model, namely CICIDS 2017 and UNSW-NB 2015. They claimed that their model achieves the accuracy rate of more than 90% while using only up to 50% of features.

The related works demonstrate a range of Intrusion Detection System implementation strategies. Most of these suggested works test their models using the datasets KDDCUP and NSL-KDD. These datasets do not, however, exhibit a lot of attribute variability. We selected the CICIDS-2017 dataset for this analysis since it has more features and dangers than the KDD dataset. To produce an effective strategy, this proposed effort concentrates on reducing the input size, or features, by using a legitimate feature optimization technique. Table 1 shows the summary of the related works discussed above.

## 3. Proposed IDS Model

To propose an IDS model, we used a deep learning algorithm called CNN with GRU. CNN is widely used as a deep learning technique in various fields. However, such techniques have long-term dependencies issues which can be addressed using GRU as discussed in the Section 1. Hence to effectively classify the attacks we used the GRU framework with RNN. Moreover, to reduce the data dimensions we pre-processed the dataset using two techniques discussed in Section 3.2.

### 3.1. Workflow

This segment manifests the flow of our proposed work. Proposed work involves two sections, namely feature optimization technique and the CNN–GRU model. Data pre-processing and classification are the two processes that form the entire workflow in Figure 1. The elimination of repetition and choosing the ideal feature set are part of the first stage. The second stage includes categorising the data and computing several parameters. The assessment of the technique’s performance in comparison with other current algorithms is shown in the subsequent part.

### 3.2. Data Pre-Processing

CICIDS 2017 dataset was used to test the effectiveness of our technique. Based on different protocols, the abstract behaviour of the 25 users was analysed. They captured the data for five consecutive days. This dataset includes various categories of attacks such as brute force, DoS/DDoS, web attacks, infiltration, botnet, port scan, etc. Table 2 displays the assault distribution in the dataset. Hence, this dataset is highly recommended to be used to test the model.

We removed the redundant instances from dataset to decrease the dimensions of dataset for classification. The dataset contained 2,300,825 instances before the removal of the redundant samples. The samples were reduced for each sub-dataset. The pattern of the same is shown in Table 3.

Moreover, by choosing the best feature subset, we employed a feature optimization strategy to lower the input dimension. The discriminative features are found in the feature set using a filter method called Pearson’s Correlation Coefficient. It determines how comparable the dataset’s features or qualities are and provides a correlation coefficient value in the [−1, 1] range. It indicates a fully positive correlation when it equals 1, and a fully negative correlation when it equals −1. This suggests that a substantial connection exists between features and a high value for the coefficient, and vice versa. The Pearson Correlation Coefficient equation is as below [26,27].
(1)$$\rho_{X,Y}=\frac{\left(X,Y\right)}{\sigma_{X}\sigma_{Y}}$$

Which can be derived as
$$=\frac{E\left(\left(X-{µ}_{X}\right)\left(Y-{µ}_{Y}\right)\right)}{\sigma_{X}\sigma_{Y}}$$
$$=\frac{E\left(XY\right)-E\left(X\right)E\left(Y\right)}{\sqrt{E\left(X^{2}\right)-E^{2}\left(X\right)\sqrt{B\left(Y^{2}\right)-E^{2}\left(Y\right)}}}$$
where ‘(X,Y)’ is a covariance measure for ‘*X*’ and ‘*Y*’, ‘$\sigma_{X}$’ and ‘$\sigma_{Y}$’ are the standard deviations for ‘*X*’ and ‘*Y*’ respectively, ‘E(X)’ is the expected value of ‘*E*’.

The dataset’s feature subsets calculated using the Pearson Correlation Coefficient equation are displayed in Table 4. Each subset consists of about 40 features out of the original dataset’s total of 77 features. To choose these attributes, a smaller set of samples without duplicate or redundant instances is also used. Pre-processing the original dataset makes it simpler and more effective when used with the suggested model.

### 3.3. Gated Recurrent Unit (GRU)

GRU is a new structure made to address the vanishing/exploding gradient problem. The upgraded LSTM framework is called GRU. For regulating the information flow, GRUs also has a gate structure similar to that of an LSTM. However, unlike LSTM, GRU lacks an output gate, allowing the content to be fully exposed. The reset and update gates are the only two gates in the GRU. The input and forget gates of the LSTM framework are combined in the second gate. Compared with LSTM, GRUs have a simpler structure and fewer parameters, which improves performance. The GRU framework has the following structure in Figure 2.

The equations below provide the GRU formulation.
(2)$$r_{t}=sigm\left(W_{xr}x_{t}+W_{hr}h_{t-1}+b_{r}\right)$$
(3)$$z_{t}=sigm\left(W_{xz}x_{t}+W_{hz}h_{t-1}+b_{z}\right)$$
(4)$$\tilde{h_{t}}=tanh\left(W_{xh}x_{t}+W_{hh}\left(r_{t}⊙h_{t-1}\right)+b_{h}\right)$$
(5)$$h_{t}=z_{t}⊙h_{t-1}+\left(1-z_{t}\right)⊙\tilde{h_{t}})$$
where ‘$x_{t}$’, ‘$h_{t}$*’, ‘*$r_{t}$’, and ‘$z_{t}$’ are the input and output vectors, reset and update gates respectively. Similar to LSTM, ‘*b*’ stands for biases and ‘*W*’ for weight, while sigmoid and tangent have the functions ‘*sigm*’ and ‘*tanh*’, respectively, for activation. Both LSTM and GRU can manage the longer dependencies. However, in terms of performance, there are some variations. In this study, we employed both frameworks to evaluate how well they classified network traffic [28,29].

### 3.4. Convolution Neural Network (CNN)

The basic architecture of CNN includes three main components, namely, convolutional, pooling, and output layer. The pooling layer is elective. The classic CNN structure having three convolution layers is widely used in image classification. It contains one input layer, several hidden layers (hidden layers include convolutional, pooling, and normalization) and a layer entirely connected to the last layer called the output layer. The neurons in one layer communicate with those in layers next to it. To reduce the proportions of the input, pooling and sub-sampling processes are executed. CNN classifier receives the input images as a group of small sub-sections which are called receptive fields. The response to the following layer is calculated with the help of mathematical convolution operations of the first or the input layer [30]. We modified the CNN architecture using the framework called GRU which is discussed in the previous section. Table 5 shows the complete architecture and strategy used in the proposed approach. The basic structure of CNN is shown in Figure 3.

As shown in Table 5, the CNN–GRU model involves three convolution layers (C) followed by two GRU layers (G) and one hidden layer (H). We used 32 neurons for the C layers and 64 neurons for the G layers. The neurons for the last layer (i.e., the output layer) are equivalent to the labels in the dataset. G and C layers use the ReLU function for the activation. The rectified linear activation function, or ReLU, output zero if the input is negative and the input directly if it is positive. The soft-max (𝑆_𝑚𝑎𝑥) activation function, which provides the output in terms of prediction probabilities, is used in the final layer. With the use of Softmax, a vector of numbers can be converted into a vector of probabilities, where each value’s probability is inversely proportional to its relative scale. The detailed steps are shown in the Algorithm 1.
**Algorithm 1.** Proposed Model**Feature Optimization and CNN–GRU****Input:**
      Data instances
**Output:**
      Confusion Matrix
      (Accuracy, precision, recall, FPR, TPR)
**Dataset Optimization**
      Remove the redundant instances
  **Feature Selection**
Using Pearson’s Correlation equation, compute the correlation of the attribute set Set $C_{f}$.
          **if** corr_value > 0.8
                add attribute to $C_{f}$
**else**
                increment in an attribute set C
**return** $C_{f}$
**Classification**
      Create training and testing sets from the dataset.
      Training set: 67%
      Testing set: 33%
      **add model**
            three Convolution layers (activation = ‘relu’)
            two GRU layers (activation = ‘relu’)
      **model compilation**
            loss function: ‘categorical_crossentropy’
            optimizer=‘adagrad’
      training CNN–GRU technique with training instances
      employing techniques to test instances
**return** Confusion Matrix $C_{m*m}$


## 4. Findings and Discussion

This section includes the assessment of the proposed technique along with a discussion of the parameters used to test its performance. In the dataset, the confusion matrix is computed for each sub-dataset. The observations of the same are discussed in the later section called performance analysis.

### 4.1. Evaluation Metrics

For each sub-dataset, we considered a number of performance metrics to test the technique. The confusion matrix is the method’s final output. A confusion matrix is an effective tool for analysing how exactly the model locates occurrences of various labels. A confusion matrix is sometimes referred to as a model’s performance summary. The confusion matrix diagram is presented in Figure 4. Precision and recall are the most often used metrics for gauging the effectiveness of DL models. Precision is the ratio of the technique’s correct or wrong estimates, whereas recall refers to the percentage of the total number of true matches and the total number of positive matches [31]. These metrics are showcased in the form of equations as follows [32,33].
(6)$$\mathrm{Precision}=\frac{\mathrm{TP}}{\left(\mathrm{TP}+\mathrm{FP}\right)}$$
(7)$$\mathrm{Recall}=\frac{\mathrm{TP}}{\left(\mathrm{TP}+\mathrm{FN}\right)}$$
(8)$$\mathrm{TPR}=\frac{\mathrm{TP}}{\left(\mathrm{TP}+\mathrm{FN}\right)}$$
(9)$$\mathrm{FPR}=\frac{\mathrm{FP}}{\left(\mathrm{FP}+\mathrm{FN}\right)}$$
(10)$$\mathrm{Accuracy}=\frac{\left(\mathrm{TP}+\mathrm{TN}\right)}{\left(\mathrm{TP}+\mathrm{TN}+\mathrm{FP}+\mathrm{FN}\right)}$$

True Positive, False Positive, False Negative, and True Negative, respectively, are represented by the abbreviations TP, FP, FN, and TN.

Moreover, True Positive Rate (TPR) and False Positive Rate (FPR) are also used to analyse the model performance. The TPR measures how many actual correct instances there are in the correct matches. The FPR counts the percentage of mismatched events among non-target occurrences. The ideal model, for instance, generates a TPR of 100% and an FPR of 0%. The TPR and FPR formulae are provided in Equations (8) and (9) correspondingly. Effectiveness of the method is then evaluated using accuracy. It is determined by applying Equation (10) to calculate the fraction of cases that the model properly classified [34,35].

### 4.2. Performance Analysis

Python programming was used to conduct the experiments on the Google Colab Cloud Environment. The dataset’s confusion matrix serves as the test’s output. Figure 5, Figure 6, Figure 7, Figure 8, Figure 9, Figure 10 and Figure 11 display the confusion matrix for each subset of the entire dataset. Table 6 through 12 display several parameters that were produced from the confusion matrix that was explained in Section 4.1. The whole dataset is distributed into 67% of the training data and 33% of the testing data.

The dataset contains over two million samples and 77 attributes. The model performed better when redundant samples and correlated features were removed from the dataset. The confusion matrix is used to determine the testing measures. The rows correspond to the true samples of the labels, while the columns of the matrix show the expected samples of the classes. Values are found diagonally in the matrix, i.e., the True Positive (TP) metric, of the brute force dataset (137,150, 1947, and 968). As more samples are examined, the model’s high classification rate becomes clearer. True negative samples, however, are samples that are incorrectly classified. Similar to this, other metrics are computed using the corresponding formulas described in Section 4.1 from the confusion matrix.

Tuesday sub-dataset’s statistical parameter is displayed in Figure 5 and Table 6. It demonstrates the predicted attack types, FTP-Patator and SSH-Patator. For each class, it was noted that our technique acquired a high True Positive Rate and precision value. Figure 6 and Table 7 display the analysis of DoS/DDoS attack, which is the biggest sub-dataset among other sub-datasets. In comparison with other sub-datasets, it also experiences the most attacks. Slowhttptest, GoldenEye, slowloris, Heartbleed and Hulk are the predicted attacks. The attack Hulk is seen to have the highest True Positive Rate, but the model fails to classify any Heartbleed attack samples properly. The statistical information from the Thursday morning sub-dataset is displayed in Figure 7 and Table 8. It displays prediction of XSS, SQL Injection, and brute force attacks. Compared with SQL injection and XSS assaults, the brute force attack was predicted more precisely. Statistical properties of the Thursday and Friday sub-datasets are displayed in Figure 8, Figure 9, Figure 10 and Figure 11 and Table 9, Table 10, Table 11 and Table 12. These reports demonstrate the prediction of attacks including infiltration, bot, DDoS, and port scans. Our approach appears to perform well when predicting Bot, DDoS, and PortScan. It struggles while trying to foresee the Infiltration, though.

The calculation of accuracies for each sub-dataset in the whole dataset is shown in Table 13. Multiple epoch settings were used to calculate each sub-accuracy dataset while taking the loss of training/validation data into account. For all subsets, it was noted that the model effectively attained greater than 95% accuracy. The Tuesday sub-dataset has the highest accuracy compared with the other subsets. Moreover, we also compared the proposed model with other Convolution Neural Network models. It was reported that the suggested model CNN–GRU achieves a classification accuracy score of 98.73% while using just about half (i.e., 58%) of the total features compared with other models. The proposed model can achieve this accuracy for the full CICIDS 2017 dataset. It also detects attacks such as SQL injection and SSH Patator which other models fail to detect. However, it fails to detect Heartbleed and XSS attacks.

Table 14 shows the assessment of the proposed technique with other existing CNN models. The accuracy graph is also shown in Figure 12. The CNN–GRU model manages to score a better accuracy rate than models developed by Samson H. et al., 2020 [15] and Sun P. et al., 2020 [17]. The other two CNN models (Priyanka V. et al., 2021 [20] and Maseer Z. et al., 2021 [33]) have slightly higher accuracy scores than our proposed technique. However, these two models were evaluated on the partial dataset. Moreover, we analysed the classification time for each sub-datasets against their sizes and accuracies respectively. We evaluated the techniques using different epochs ranging from 20 to 100 for different sub-datasets. As shown in Figure 13, the classification time for the model is high when the size of the sub-dataset is more. The DoS/DDoS sub-dataset has the highest size among all the sub-datasets. Hence the time taken by the model to classify DoS/DDoS attacks is more compared with other sub-datasets. However, this trend is not the same for all the sub-datasets. Even though their sizes are smaller than those of other sub-datasets, the DDoS and WebAttacks sub-dataset classification times are longer. Figure 14 shows the analysis of the classification time with respect to the accuracy score. The accuracy of the DoS/DDoS sub-dataset is less compared with other sub-datasets. However, due to its size the classification time for the same is higher.

To achieve optimum accuracy and minimal loss for each sub-dataset in this study, we used a varied number of epochs. The distribution of epochs for the sub-datasets is shown in Table 15. The loss is a result of the model’s inaccurate label predictions. We calculated the loss in our proposed technique using the Sparse Categorical Cross-Entropy technique. The epoch vs accuracy and loss graphs are shown in Figure 15, Figure 16, Figure 17, Figure 18, Figure 19, Figure 20 and Figure 21.

## 5. Conclusions

In this research, we designed a DL technique to develop an IDS. We used CNN with the GRU framework. We used three CNN layers and two GRU sequences. Both sequences use an activation function called ReLU to define the output of the provided inputs. Following that, the concatenate merge mode is used to combine the outputs of both sequences. The dataset considered in this work offers a diversity of attacks and a high number of instances, which was used to verify the suggested model. The dataset contains network data recorded from eight different sessions. In this study, we considered seven sessions and overlooked one session which contains just normal traffic data. We optimized the original dataset by selecting features based on Pearson’s Correlation coefficient. We also removed redundant or duplicate instances from the dataset. The evaluation of the model was performed using measurement parameters of the confusion matrix. The suggested model achieved a low FP rate and a classification accuracy rate of 98.73%. The technique used nearly half of the total attributes (less than 58% of the total for each sub-dataset) compared with all other classifiers.

## 6. Future Scope

The proposed method performed well, but by optimizing the technique even further, performance can be improved. A few attacks were unsuccessfully categorised using the proposed approach. One of the reasons is the imbalanced data of attacks. Therefore, for all the attacks in the dataset, we strive for highly optimal training data in future study. Most classical physical-layer security techniques require CSI, but it is hard to obtain due to time-varying wireless declining [37,38]. As one of the most serious cyber attacks, APT caused global concern. APT is a persistent, multi-stage attack that aims to compromise a system and gain information from it, causing damage and financial loss [39]. The Digital Twin, biometric system, and CPS usher in a new era for commerce, particularly in the health sector, by tracking individuals’ health data in order to provide needful, fast, and efficient services to users [40,41]. IDSs are configured so that they learn from historical network traffic data and detect both normal and abnormal event connections from the monitored system. However, due to the massive amount of historical data, this system may suffer from issues such as accuracy, false alarms, and execution time [42].

## Figures and Tables

**Figure 1 sensors-23-00890-f001:**
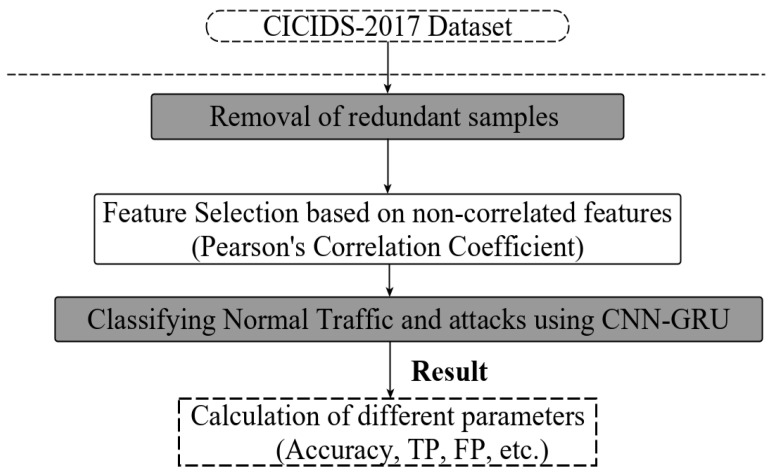
Proposed model layout.

**Figure 2 sensors-23-00890-f002:**
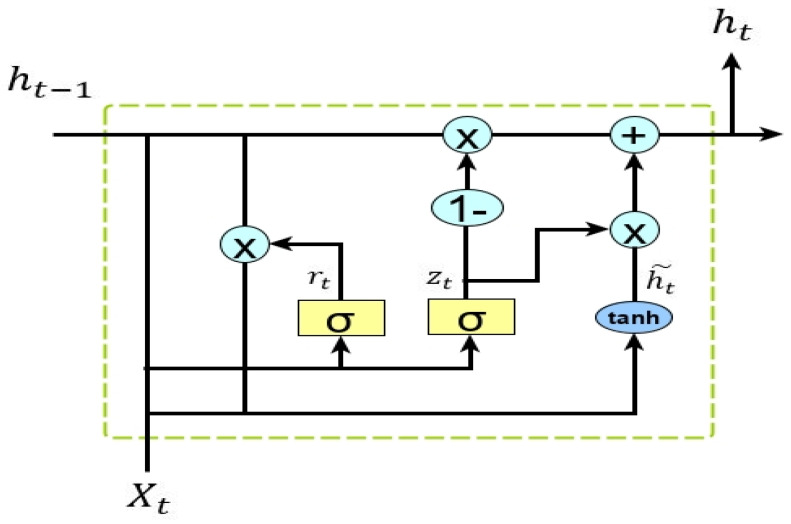
Architecture of GRU.

**Figure 3 sensors-23-00890-f003:**
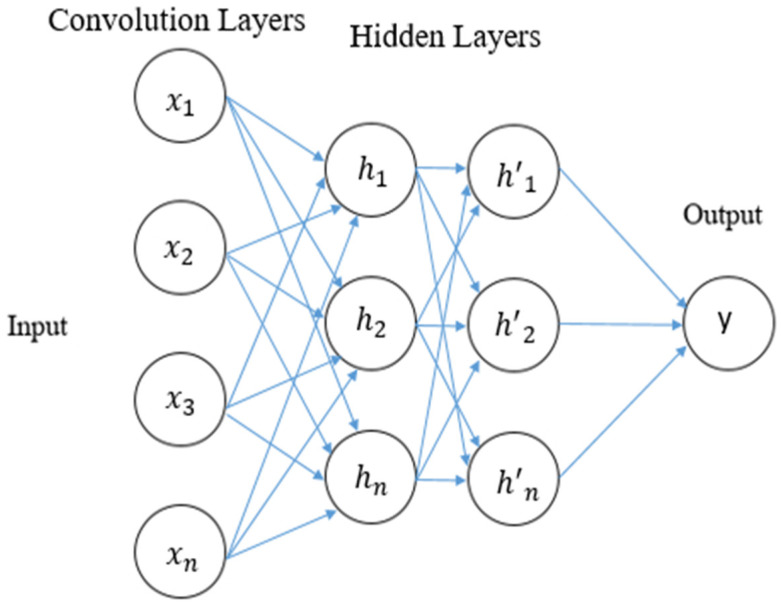
The architecture of CNN.

**Figure 4 sensors-23-00890-f004:**
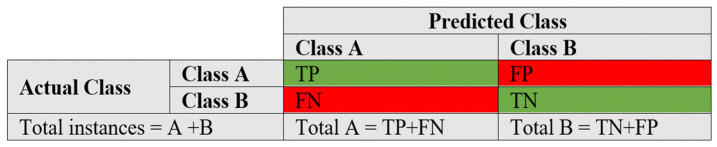
Confusion matrix.

**Figure 5 sensors-23-00890-f005:**
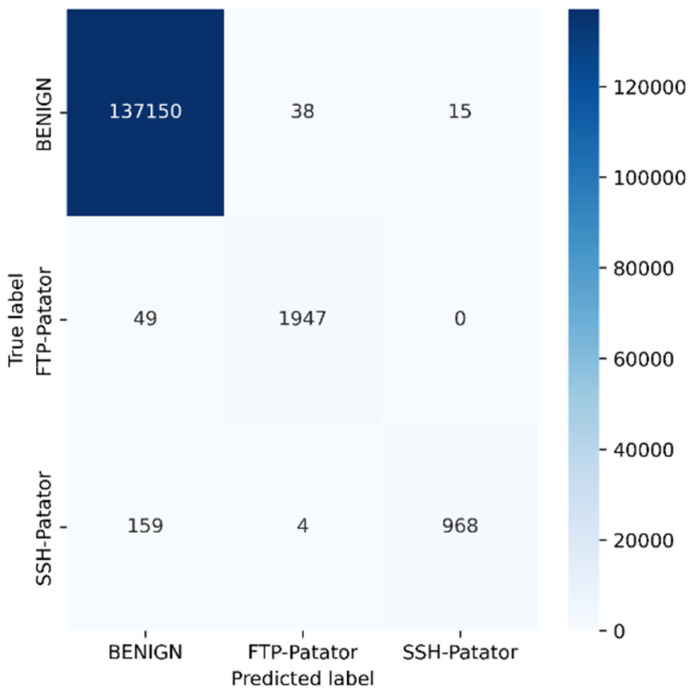
Brute force.

**Figure 6 sensors-23-00890-f006:**
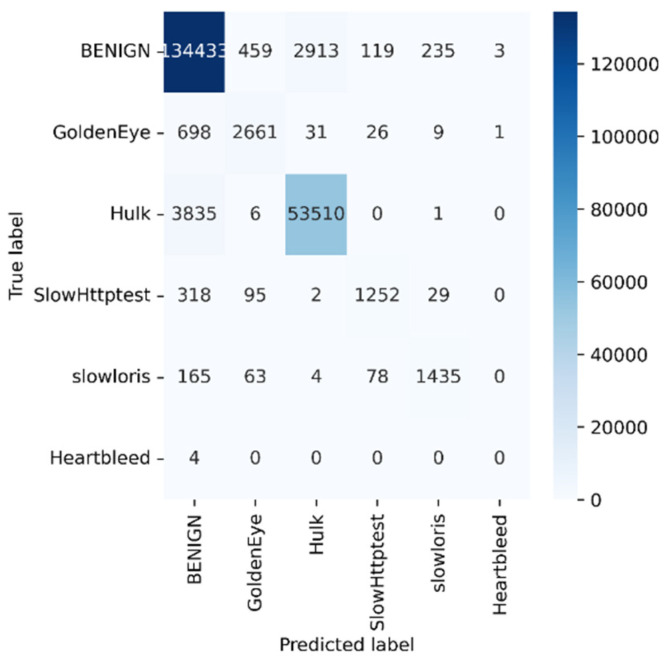
DoS/DDoS.

**Figure 7 sensors-23-00890-f007:**
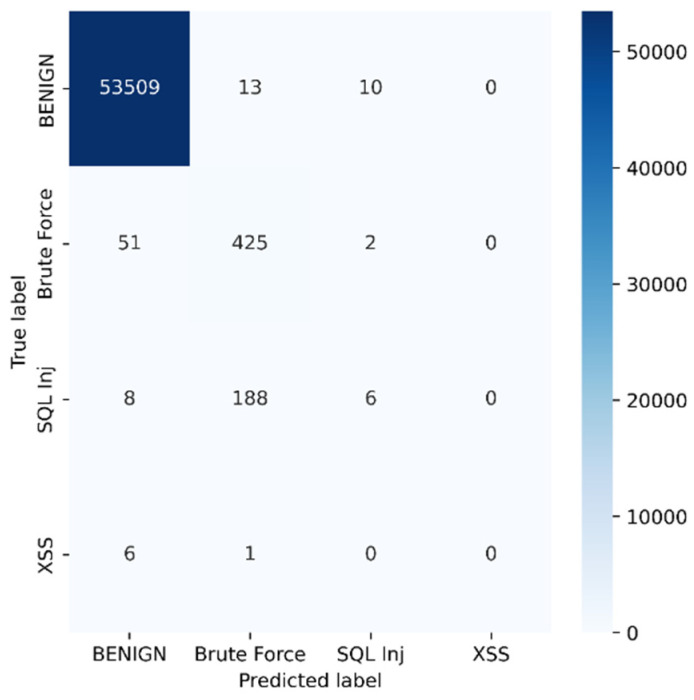
Web attacks.

**Figure 8 sensors-23-00890-f008:**
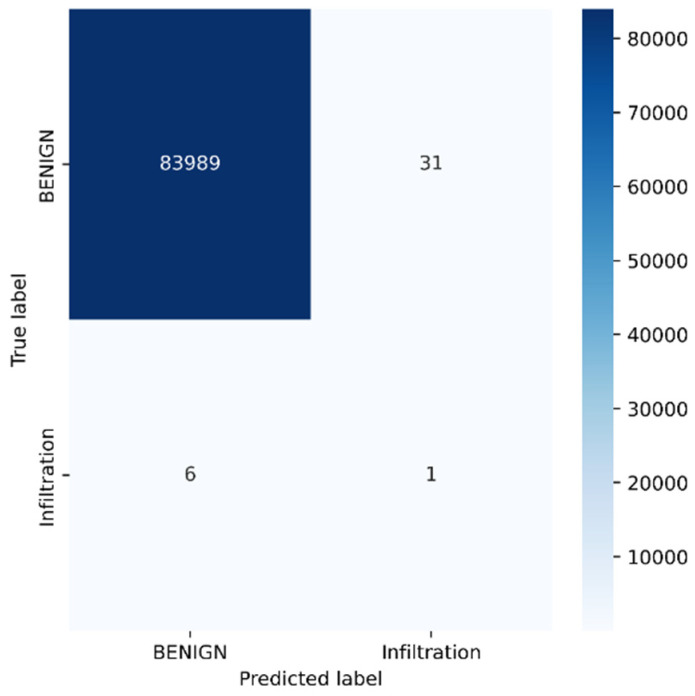
Infiltration.

**Figure 9 sensors-23-00890-f009:**
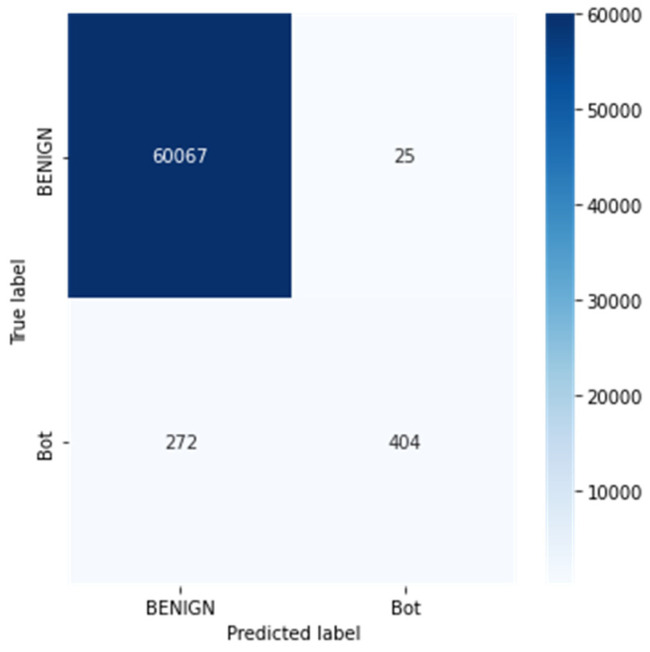
Bot.

**Figure 10 sensors-23-00890-f010:**
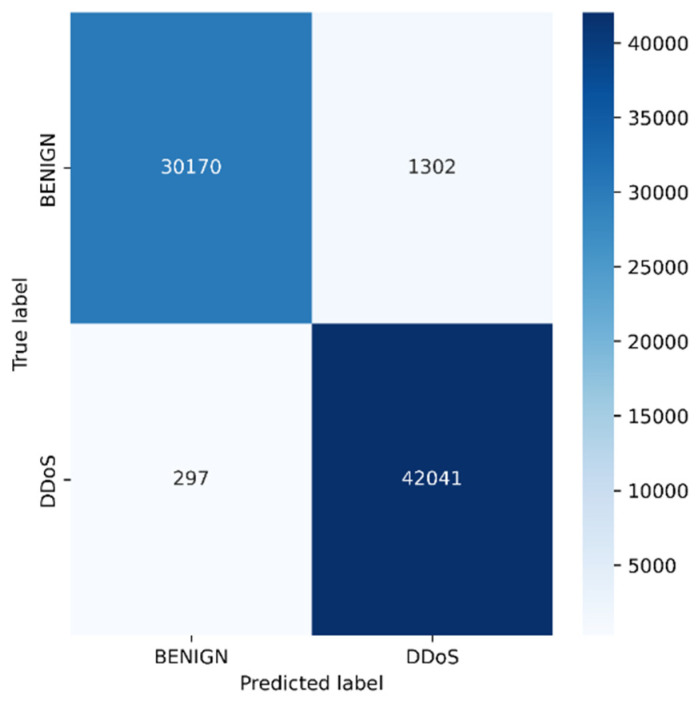
DDoS.

**Figure 11 sensors-23-00890-f011:**
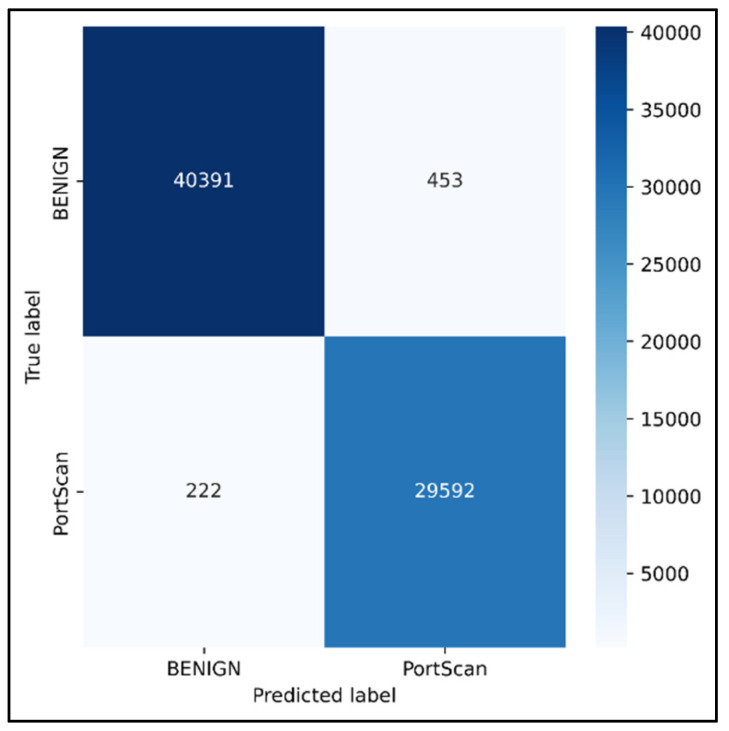
PortScan.

**Figure 12 sensors-23-00890-f012:**
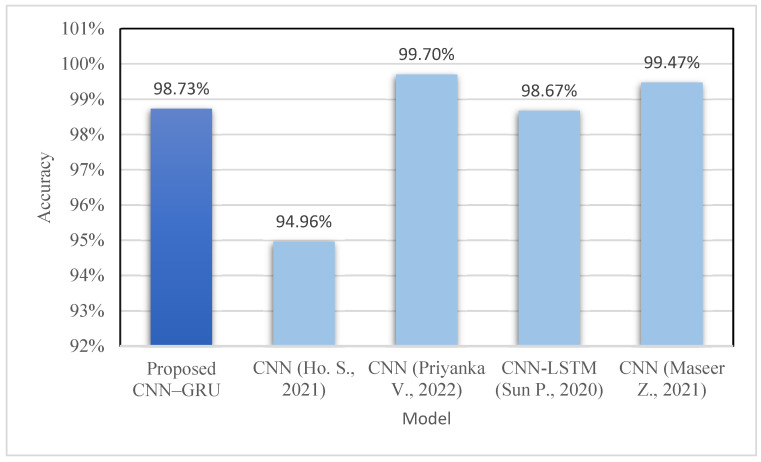
Accuracy graph [18,19,20,36].

**Figure 13 sensors-23-00890-f013:**
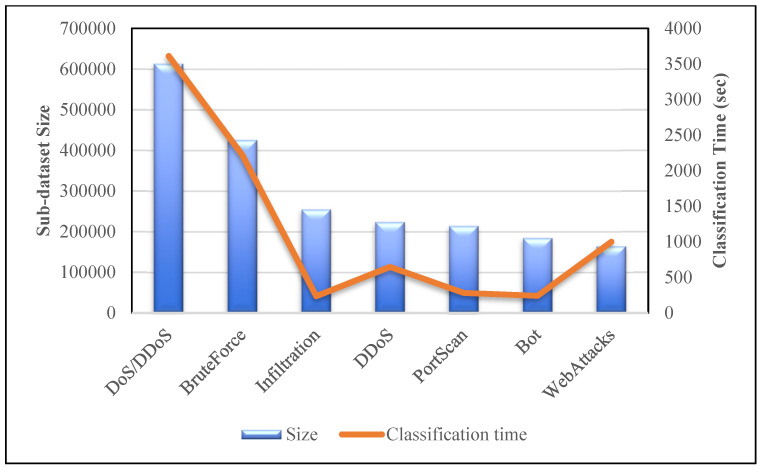
Dataset size vs. classification time.

**Figure 14 sensors-23-00890-f014:**
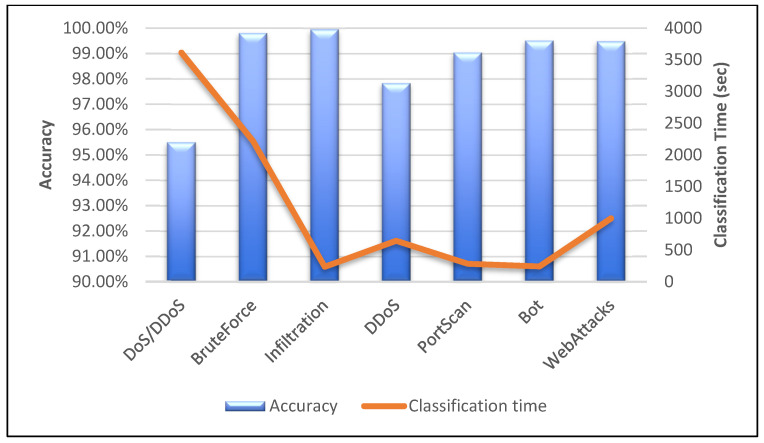
Accuracy vs. classification time.

**Figure 15 sensors-23-00890-f015:**
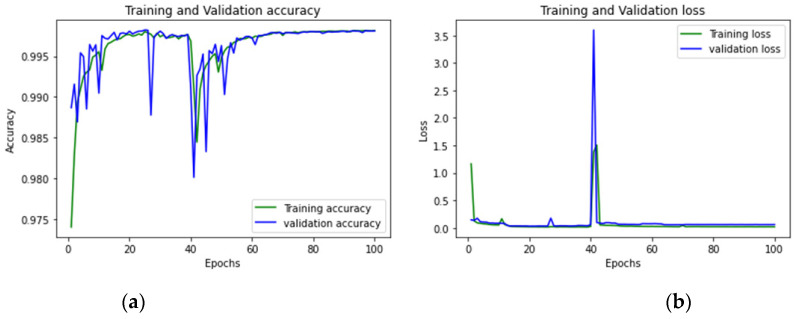
(**a**,**b**): Tuesday: Accuracy curve and Loss curve.

**Figure 16 sensors-23-00890-f016:**
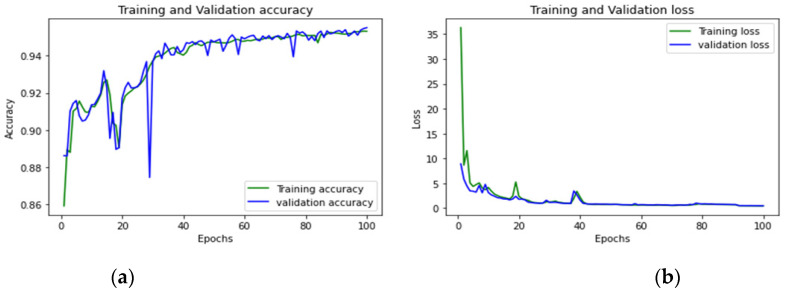
(**a**,**b**): Wednesday: Accuracy curve and Loss curve.

**Figure 17 sensors-23-00890-f017:**
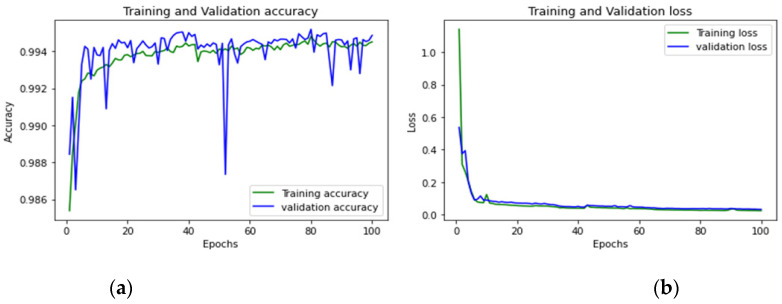
(**a**,**b**): Thursday morning: Accuracy curve and Loss curve.

**Figure 18 sensors-23-00890-f018:**
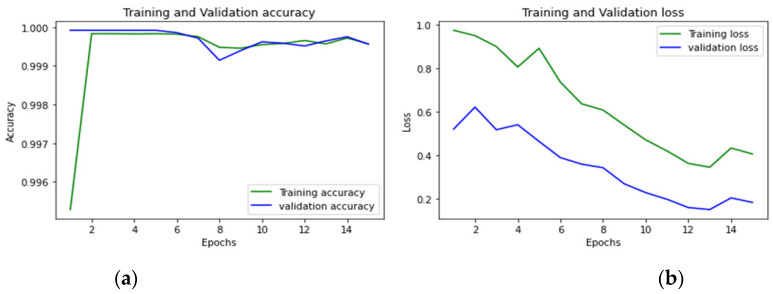
(**a**,**b**): Thursday afternoon: Accuracy curve and Loss curve.

**Figure 19 sensors-23-00890-f019:**
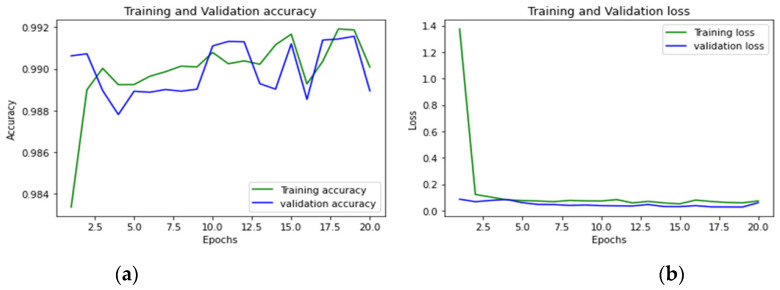
(**a**,**b**): Friday morning: Accuracy curve and Loss curve.

**Figure 20 sensors-23-00890-f020:**
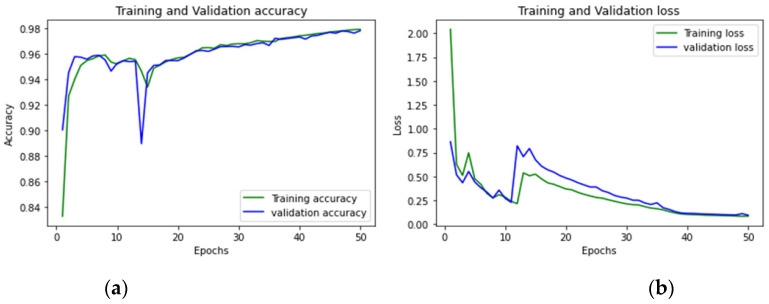
(**a**,**b**): Friday afternoon-DDoS: Accuracy curve and Loss curve.

**Figure 21 sensors-23-00890-f021:**
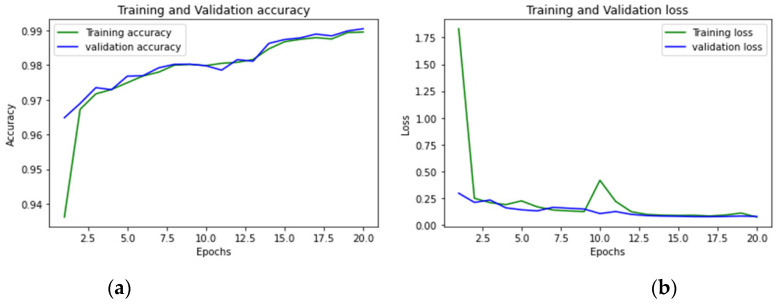
(**a**,**b**): Friday afternoon PortScan: Accuracy curve and Loss curve.

**Table 1 sensors-23-00890-t001:** Summary of the related work.

Reference	Existing Technique	Summary
Roberto et al., (2020) [10]	NIDS -ML Approaches	Classifier: Logistic regression (LR), SVC-L, SVC-RBF, RF Dataset: UGR’16 Metrics: Precision, Recall, F1, AUC Programming: Python, scikit-optimize
Amar et al., (2020) [11]	Hybrid IDS using ML	Classifier: KNN, NB, LR, SVM Dataset: NSL-KDD Metrics: Accuracy, DR, FAR, Precision, Recall, F1
Iram et al., (2020) [12]	An ML Approach for IDS on NSL-KDD Dataset	Classifier: SVM, KNN, LR, NB, MLP, RF, ETC, DT Dataset: NSL-KDD Metrics: Precision, Recall, Accuracy, F1
Abdulsalam et al., (2021) [13]	NIDS Based on Machine Learning for SDNs	Classifier: DT, RF and XGBoost Dataset: NSL-KDD Metrics: Accuracy, DR, ROC, F-score, Precision, Recall
Raisa et al., (2022) [14]	IDS using GIWRF feature selection technique	Classifier: DT, GBT, MLP, AdaBoost, LSTM, GRU Dataset: UNSW-NB 15, Network ON_IoT Metrics: Precision, Recall, F1, FPR Programming: Python (Jupyter Notebook)
Achmad et al., (2021) [15]	Hybrid ML method for increasing the performance of network IDS	Classifier: Decision Tree Dataset: NSL-KDD, UNSW-NB15 Metrics: Accuracy, sensitivity, specificity, false alarm rate Programming: Python (Jupyter Notebook)
Gustavo et al., (2021) [16]	An End-to-End Framework for ML-Based NIDS	Classifier: KNN, RF, XGB, NB, DT, MLP, SVM, and LR Dataset: proposed dataset, MAWILab dataset Metrics: F1, FPR, precision, recall, storage, time, CPU and RAM usage Programming: Python (scikit-learn)
Maonan et al., (2020) [17]	An explainable ML Framework for IDS	Classifier: KNN, RF, SVM-RBF, one-vs.-all, and multiclass classifier Dataset: NSL-KDD Metrics: Accuracy, F1, FPR, precision, recall, connections between certain characteristics and attack types Programming: Python (Pytorch)
Samson et al., (2021) [18]	Detection of Known, Innovative Cyberattacks Using CNN	Classifier: Convolution Neural Network CNN Dataset: CICIDS 2017 Metrics: TNR, DR, accuracy, FPR
Priyanka et al., (2020) [19]	Performance Assessment of IDS-CICIDS-2017 Dataset	Classifier: RF, NB, CNN Dataset: CICIDS2017 (partial) Metrics: Precision, Recall, F1, accuracy Programming: Python (Jupyter Notebook)
Sun et al., (2020) [20]	CNN-LSTM hybrid network	Classifier: CNN-LSTM Dataset: CICIDS 2017 Metrics: Accuracy, TPR, FPR, F1 Programming: Python
Mario et al., (2020) [21]	Neural-based approaches for Network Intrusion Management	Classifier: ANN Dataset: CICIDS2017/2018 and KDD99 Metrics: Accuracy, f-measure, precision, recall, time complexity Programming: Python
Shi et al., (2021) [22]	Semi-Supervised Deep Reinforcement Learning	Classifier: SSDDQN Dataset: NSL-KDD and AWID Metrics: Accuracy, precision, recall, DR, FPR, efficiency Programming: Python
Charlotte et al., (2019) [23]	DL for the Classification of Sentinel-2	Classifier: RF, RNN, CNN Dataset: Sentinel-2 images Metrics: Accuracy, runtime Programming: Python
Joohwa et al., (2020) [24]	NIDS using Deep Sparse Autoencoder	Classifier: Single RF, DSAE-RF Dataset: CICIDS 2017 Metrics: Accuracy, precision, F1 Programming: Python
Mohammadnoor et al., (2020) [25]	Multi-Stage Optimized ML for NIDS	Classifier: KNN, RF Dataset: CICIDS 2017 and the UNSW-NB 2015 Metrics: Accuracy, precision, recall, FAR Programming: Python

**Table 2 sensors-23-00890-t002:** Dataset attacks.

Sub-Dataset	Attacks
Tuesday Samples	benign ftpPatatorAttack sshPatatorAttack
Wed. Samples	benign goldeneyeAttack hulkAttack slowhttptestAttack slowlorisAttack heartbleedAttack
Thur. Morning Samples	benign bruteForceAttack SqlInjectionAttack XSSAttack
Thurs. Afternoon Samples	benign infiltrationAttack
Fri. Morning Samples	benign botAttack
Fri. Afternoon Samples-DDoS	benign ddosAttack
Friday Afternoon Samples-PortScan	benign portscanAttack

**Table 3 sensors-23-00890-t003:** Distribution of data in the dataset.

Sub-Dataset	Number of Instances	
	With Redundancy	Without Redundancy
Tuesday_BruteForce	445,909	425,240
Wednesday_DoS/DDoS	692,703	613,287
Thurs_Morning_WebAttacks	170,366	164,300
Thurs. Afternoon_Infiltration	288,602	254,625
Fri. Morning_Bot	191,033	184,145
Fri. Afternoon-DDoS	225,745	223,666
Fri. Afternoon-PortScan	286,467	214,114

**Table 4 sensors-23-00890-t004:** Selected attributes.

Sub-Dataset	Total Features	Selected Features
BruteForce	77	43
DoS/DDoS	77	41
WebAttacks	77	39
Infiltration	77	40
Bot	77	37
DDoS	77	39
PortScan	77	37

**Table 5 sensors-23-00890-t005:** Model strategy.

Parameter	1	2	3	4	5	6
Type	C	C	C	G	G	H
No. of neurons	32	32	32	64	64	No. of classes
Filter Size	(1)	(1)	(1)	(1)	(1)	(1)
Activation Fun.	ReLU	ReLU	ReLU	ReLU	ReLU	S_max_

**Table 6 sensors-23-00890-t006:** Statistical analysis of brute force.

Parameters	BENIGN	FTPPatator	SSHPatator
Precision	0.9984	0.9788	0.9847
Recall	1.00	0.98	0.86
TN	2919	138,292	139,184
FP	208	42	15
TP	137,150	1947	968
FN	53	49	163
FPR	0.0665	0.0003	0.0001
TPR	0.9996	0.9754	0.8558

**Table 7 sensors-23-00890-t007:** Statistical analysis of DoS/DDoS.

Parameters	BENIGN	Goldeneye	Hulk	Slowhttptest	Slowloris	Heartbleed
Precision	0.9640	0.8102	0.9477	0.8488	0.8396	0
Recall	0.97	0.78	0.93	0.74	0.82	0
TN	59,203	198,336	142,083	200,466	200,366	202,377
FP	5020	623	2950	223	274	4
TP	134,433	2661	53,510	1252	1435	0
FN	3729	765	3842	444	310	4
FPR	0.078	0.003	0.020	0.001	0.001	0
TPR	0.9730	0.7767	0.9330	0.7382	0.8223	0

**Table 8 sensors-23-00890-t008:** Statistical analysis of Web attacks.

Parameters	BENIGN	Brute Force	Sql Inj	XSS
Precision	0.9987	0.6778	0.3333	0
Recall	1.00	0.89	0.03	0.00
TN	622	53,539	54,005	54,212
FP	65	202	12	0
TP	53,509	425	6	0
FN	23	53	196	7
FPR	0.0946	0.0037	0.0002	0
TPR	0.9995	0.8891	0.0297	0

**Table 9 sensors-23-00890-t009:** Statistical analysis of infiltration.

Parameters	BENIGN	Infiltration
Precision	0.9999	0.0312
Recall	1.00	0.14
TN	1	83,989
FP	6	31
TP	83,989	1
FN	31	6
FPR	0.8571	0.0003
TPR	0.9996	0.1428

**Table 10 sensors-23-00890-t010:** Statistical analysis of Bot.

Parameters	BENIGN	Bot
Precision	0.9954	0.9417
Recall	1.00	0.60
TN	404	60,067
FP	272	25
TP	60,067	404
FN	25	272
FPR	0.4023	0.0004
TPR	0.9995	0.5976

**Table 11 sensors-23-00890-t011:** Statistical analysis of DDoS.

Parameters	BENIGN	DDoS
Precision	0.9902	0.9699
Recall	0.96	0.99
TN	42,041	30,170
FP	297	1302
TP	30,170	42,041
FN	1302	297
FPR	0.0070	0.0413
TPR	0.9586	0.9929

**Table 12 sensors-23-00890-t012:** Statistical analysis of PortScan.

Parameters	BENIGN	PortScan
Precision	0.9945	0.9849
Recall	0.99	0.99
TN	29,592	40,391
FP	222	453
TP	40,391	29,592
FN	453	222
FPR	0.0074	0.0110
TPR	0.9889	0.9925

**Table 13 sensors-23-00890-t013:** Accuracy of sub-datasets.

Sub-Dataset	Accuracy
BruteForce	99.81%
DoS/DDoS	95.50%
WebAttacks	99.48%
Infiltration	99.95%
Bot	99.51%
DDoS	97.83%
PortScan	99.04%

**Table 14 sensors-23-00890-t014:** Performance of models.

Model	No. of Attributes	Accuracy
Proposed CNN–GRU	Less than 44	98.73%
CNN [18]	77	94.96%
CNN [19]	77	99.7%
CNN-LSTM [20]	77	98.67%
CNN [36]	77	99.47%

**Table 15 sensors-23-00890-t015:** Epochs.

Sub-Dataset	Epochs
BruteForce	100
DoS/DDoS	100
WebAttacks	100
Infiltration	15
Bot	20
DDoS	50
PortScan	20

## Data Availability

No data are associated with this research work.

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
