# Peer review of "Composition of Hybrid Deep Learning Model and Feature Optimization for Intrusion Detection System"

_sensors, 2023, doi:10.3390/s23020890_

Round 1

Reviewer 1 Report

In this paper, the authors propose a technique which combines both Convolutional Neural Networks (CNN), and Gated Recurrent Unit (GRU), where different CNN-GRU combination sequences are presented to optimize the network parameters. Paper is clearly written, logically organized, very comprehensive and impressive amount of research and study has been carried out. The content is technically sound and contains sufficient interest.

The reviewer has some concerns.

·       The paper lacks in-depth discussions in section "Performance Analysis"

·       The quality of all figures should be improved.

·  Please improve the reference format and verify the number of each reference cited in the paper

·    Some sentences are too long to make readers confused, and there are also some typos and grammar errors in this paper.

Author Response

The rebuttal for reviewer 1 comments are attached.

Reviewer 2 Report

In this manuscript, the authors propose an IDS framework to classify attacks relying on a combination of deep learning techniques including CNNs and GRUs. As a reference dataset, the authors consider the CIC-IDS2017 which provides a good dataset to test the technique. 

The first concern is that the authors must make an effort to better convince the Reader about the novelty of their proposal since the usage of deep learning techniques in the realm of intrusion detection systems appears to be very inflated.

The second concern pertains to the performance analysis that is not adequately complemented with a time complexity analysis. 

The complexity introduced by some structures (e.g., convolutional layers used by CNNs), in fact, could negatively affect the time complexity. In this connection, the authors could gather data about the time consumed by each technique (in addition to the classic accuracy analysis). At this aim, the related work section could be enriched by including some credited works that consider time consumed by each technique beyond the accuracy/performance evaluation:

- Experimental Review of Neural-Based Approaches for Network Intrusion Management," in IEEE Transactions on Network and Service Management, 2020;

- Network Abnormal Traffic Detection Model Based on Semi-Supervised Deep Reinforcement Learning, in IEEE Transactions on Network and Service Management, 2021;

- Deep Learning for the Classification of Sentinel-2 Image Time Series, IEEE IGARSS, 2019.

In addition, since the pre-processing feature selection stage lacks time complexity analysis as well, the authors should consider discussing such a point.

Moreover, some related works aimed at analyzing the impact of the feature extraction/selection process on network intrusion detection are missing. For instance:

- "Network Intrusion Detection System using Feature Extraction based on Deep Sparse Autoencoder," (IEEE ICTC conference, 2020);

- "Multi-Stage Optimized Machine Learning Framework for Network Intrusion Detection", (IEEE ,Trans. on Netw. and Serv. Management, 2020);

Some minor concerns:

- Most of the figures appear to be very blurry, thus, I suggest to improve their quality.

Author Response

The rebuttal of Reviewer 2 is attached

Round 2

Reviewer 2 Report

The authors have addressed all the comments raised in the previous round of review. In particular, they have:

- better stress the novelty of their proposal;

- complemented performance analysis with a time complexity analysis

- updated the pertinent technical literature

In my opinion, the paper can be accepted in its current form.